# Ultrasound-Based Morphological and Functional Assessment in Male CrossFit Athletes with Unilateral Subacromial Shoulder Pain: An Observational Study

**DOI:** 10.3390/medicina61071304

**Published:** 2025-07-19

**Authors:** Fabien Guerineau, Ann Cools, Jaime Almazán-Polo, María Dolores Sosa-Reina, Vanesa Abuín-Porras, Cristian Baroa-Fernández, Pablo García-Ginés, Ana Román-Franganillo, Ángel González-de-la-Flor

**Affiliations:** 1Department of Physiotherapy, Faculty of Medicine, Health and Sport Sciences, European University of Madrid, Villaviciosa de Odón, 28670 Madrid, Spain; fabien.guerineau@universidadeuropea.es (F.G.); mariadolores.sosa@universidadeuropea.es (M.D.S.-R.); vanesa.abuin@universidadeuropea.es (V.A.-P.); cristianbaroaf@gmail.com (C.B.-F.); pablogarciagines2@gmail.com (P.G.-G.); anaromanfranganillo@gmail.com (A.R.-F.); angel.gonzalez@universidadeuropea.es (Á.G.-d.-l.-F.); 2Department of Rehabilitation Sciences, Faculty of Health Sciences, Campus UZ Gent, Ghent University, Corneel Heymanslaan 10, B3, Entrance 46, 9000 Gent, Belgium; ann.cools@ugent.be; 3Faculty of Nursing, Physiotherapy and Podiatry, Universidad Complutense de Madrid, 28040 Madrid, Spain

**Keywords:** shoulder impingement syndrome, subacromial shoulder syndrome, shoulder pain, overhead athletes, ultrasound

## Abstract

*Background and Objectives*: CrossFit is a discipline involving a wide range of overhead movements performed at high intensity and under accumulated fatigue that predispose to a high risk of shoulder complex injuries. This study aimed to compare ultrasonographic findings between symptomatic and asymptomatic shoulders in CrossFit athletes. *Materials and Methods*: A cross-sectional study was conducted to compare ultrasound parameters between the painful and non-painful shoulders in CrossFit athletes with unilateral subacromial shoulder pain. Assessed variables included subacromial subdeltoid bursa thickness, supraspinatus tendon thickness, the acromiohumeral distance, the coracoacromial ligament distance, the bicipital groove angle, cross-sectional area of the biceps brachii longus head tendon, as well as the serratus anterior and lower trapezius muscle thickness. *Results*: Twenty male CrossFit athletes (forty shoulders) with an average age of 25.70 ± 4.03 years participated in the study. A statistically significant increase was observed (*p* < 0.05) in the subacromial subdeltoid bursa thickness in the painful shoulder compared to the asymptomatic side. All other ultrasound parameters did not show statistically significant differences. *Conclusions*: Only subacromial subdeltoid bursa thickness differed significantly between sides. This isolated finding may not fully explain shoulder pain, which cannot be solely attributed to morphological changes. Further research is needed to determine the relationship between shoulder pain and ultrasound features in CrossFit athletes, as well as the role of ultrasound in predicting structural changes in pain conditions.

## 1. Introduction

CrossFit is a sports discipline based on the combination of strength and endurance exercises. It involves constantly varied functional movements applied intermittently at high intensity, with special involvement of overhead kinetic chains [1]. Thus, CrossFit athletes are particularly predisposed to shoulder injuries, with a 28.6% prevalence rate, especially in lifting activities or gymnastics tasks [2]. Shoulder injuries are 2.79% higher in men than in women [3]. Overhead athletes are predisposed to develop subacromial shoulder pain (SSP), a condition frequently linked to mechanical overload of subacromial structures such as the supraspinatus tendon, the subacromial-subdeltoid bursa, or the coracoacromial arch. In this context, the biomechanics of repetitive overhead movements play a key role in both the onset and persistence of pain, and should therefore guide the clinical and imaging evaluation of the painful athletic shoulder [4,5,6]. Despite the difficulties in classifying shoulder pain from a pathomechanical and/or functional point of view, SSP as the “umbrella” term encompasses shoulder pain associated with glenohumeral structures typically described in overhead movements, such as the snatch or the jerk variants in CrossFit [7,8,9]. In this context, the need for a wide range of pain-free movements, combined with optimal motor control and strength, makes CrossFit a highly complex and demanding movement pattern. The considerable demand for push and pull movement patterns of the upper extremity makes this discipline functionally demanding, requiring a comprehensive evaluation that integrates biomechanical, functional, and sensorimotor components [10,11].

Considering the shoulder complex assessment, an integrated approach combining physical, functional, and psychological variables plays a fundamental role in the clinical examination of overhead athletes [12]. Recently, the development of ultrasound imaging (USI) and its accessibility poses a challenge when it comes to combining the information obtained from an imaging tool with the information obtained through the examination [13]. USI is considered a safe, non-invasive, and cost-effective tool for musculoskeletal shoulder evaluation, especially when compared to more expensive or static imaging modalities such as MRI. While MRI has been used to characterize shoulder pathologies in CrossFit athletes [14], USI provides a more accessible alternative for routine structural and functional assessment. Its dynamic capability enables the clinician to assess structural integrity and movement-related alterations in real time, which is particularly relevant in pathologies with a functional component, such as SSP. In addition, it facilitates the identification of anomalies in symptomatic shoulders with SSP, including alterations in supraspinatus tendon thickness (SST), subacromial deltoid bursa (SADB), and acromiohumeral distance (AHD). These alterations may reflect adaptations to repetitive mechanical loading, bursal irritation, or reduced subacromial space, all of which are biomechanically plausible in high-volume overhead athletes such as CrossFit practitioners [15,16]. Moreover, USI allows for the dynamic evaluation of muscle activation patterns. Changes in muscle thickness between rest and contraction have been used to examine functional behavior of stabilizing muscles such as the lower trapezius (LT) or serratus anterior (SA), which may be compromised in athletes with shoulder pain. This functional dimension enhances the diagnostic value of USI, allowing clinicians to assess both static morphology and contractile behavior under load [17,18].

Several studies have attempted to identify structural differences between individuals with and without shoulder pain, as well as between the painful shoulder (PS) and the non-painful shoulder (NPS) within the same individual [19,20]. However, to our knowledge, no previous study has conducted a bilateral comparison of both structural and functional ultrasound findings between symptomatic and asymptomatic shoulders specifically in CrossFit athletes. This is particularly relevant given the sport’s high exposure to complex overhead gestures and the documented prevalence of SSP. Thus, this study aimed to highlight the structural and functional differences identified by USI between the PS and NPS in male CrossFit athletes with unilateral SSP. We hypothesized that the painful shoulder would present measurable structural and functional differences compared with the asymptomatic shoulder.

## 2. Methods and Methods

### 2.1. Study Design

A cross-sectional investigation was designed in accordance with the guidelines of the Strengthening the Reporting of Observational Studies in Epidemiology (STROBE) statement [21]. This study aimed to assess the USI features of the shoulder joint in CrossFit athletes with symptoms of unilateral shoulder pain. The principles of the Declaration of Helsinki and all regulations pertaining to human experimentation were adhered to in this study [22]. The study was reviewed and approved by the ethics committee of Universidad Europea de Madrid under the internal code CIPI/23.011 and date of approval 30 January 2023.

### 2.2. Sample Size Calculations

The sample size was determined according to a previous study by Kjaer et al. that assessed the USI of the shoulder structures using the G*Power 3.1.9.2 software (G*Power ©, University of Dusseldorf, Düsseldorf, Germany), the reported values of the SADB variable of intra-rater reliability (ICC: 0.97), the minimum detectable change (MDC: 21%, 0.37 mm), and standard error measurement (SEM: 0.08 mm) [23]. The calculation of the effect size was carried out through the mean difference in SADB measurement plus a 20% n change (NPS: 1.71; PS: 2.15), together with the pooled SD (0.51), determining an effect size of 0.862, considering a 1:1 allocation ratio, resulting in a total sample of 36 shoulders, finally evaluating 40 shoulders of 20 participants (N = 40; PS = 20; NPS = 20; actual power 0.813).

### 2.3. Participants

Twenty male (N = 20) CrossFit athletes with unilateral shoulder pain were recruited, evaluated bilaterally (n = 40) and divided into two groups of 20 PS (n = 20) and 20 NPS (n = 20) groups. All participants were informed and gave signed consent for participation in the study. Data were collected between January and June 2023 and inclusion criteria for study enrollment were: (a) male CrossFit athletes; (b) aged 18–35; (c) with at least two years of experience in CrossFit discipline; (d) athletes who follow at least of 4 days a week of training; (e) competitive athletes classified in the Rx or Elite category, according to CrossFit’s standardized performance classification system; (f) athletes following a structured and individualized training program; and (g) presenting at least three of the following five shoulder impingement criteria: positive Neer sign, positive Hawkins’s sign, positive Jobe sign, positive painful arc nsign and positive pain in resisted external rotation in zero degrees of shoulder abduction [24]. Additionally, (h) participants had to present pain in overhead weightlifting activities in the last 3 months, such as snatch and jerk variants during training or competition. Participants were excluded if they reported any of the following criteria: a history of surgical interventions or congenital alterations; metabolic, neurological, autoimmune, or cardiovascular diseases affecting the shoulder; inability to understand the study protocol; cognitive impairments; taking analgesic or anti-inflammatory medication during the study or a week before; or a low physical activity level on the International Physical Activity Questionnaire (IPAQ) in the last week.

### 2.4. Descriptive Data

Anthropometric measurements were collected, including age (years), height (m), weight (kg), and body mass index (BMI) [25]. Participants were also asked about their dominant side and the PS.

### 2.5. USI Assessment

The USI assessment was conducted by an experienced evaluator with over five years of expertise in musculoskeletal USI assessment who was blinded to the PS of each participant. To minimize potential examiner bias, participants were instructed not to disclose their symptomatic side during evaluation. All assessments were systematically performed in the same sequence, beginning with the right shoulder followed by the left shoulder, regardless of symptom presentation. The symptomatic side was only identified during the final data analysis. All participants were evaluated in a sitting position, using ultrasound equipment (Logiq S7 Expert US, GE Healthcare, Chicago, IL, USA) with a wide-spectrum linear probe (ML6-15 H40452LY Wide-Spectrum Linear Array Probe, field of view 50 mm) with a frequency range of 4–15 MHz, and a preset system was established to standardize USI evaluation (depth, 4.5 cm, frequency, 12 MHz; fifty-five gain points; sixty-nine dynamic range points; and one focus positioned 2 cm deep). All images and videos were stored in DICOM format and analyzed using the open - source Fiji Software. (U.S. National Institutes of Health; Bethesda, MD, USA) (Figure 1) [23]. All scans and measurements were conducted three times to obtain the average of the three measurements. Although all measurements were performed by a single experienced examiner to ensure consistency, intra- and inter-rater reliability were not assessed within this study. However, the selected measurement protocols have demonstrated excellent intra-rater reliability in prior research, particularly for SADB and SST variables [26].

SST and SADB were evaluated in a seated position in the “modified Crash position” for shoulder examination (Figure 2A). The ultrasound probe was placed in the long axis, using the anterior edge of the acromion and the insertion footprint of the supraspinatus as reference, known as the parrot-beak image (Figure 2B). SST measurements were performed using the most proximal and concave points of the insertion footprint as the measurement points to the most superficial point of the tendon [26]. An SADB thickness measurement was performed using a perpendicular line 20 mm from the SST insertion point on the humerus [26].

In the evaluation of the AHD, the patient’s arm was positioned in a neutral position, with the elbow flexed at 90°, and a cushion beneath the participant’s elbow to eliminate resistance from the weight. The probe was positioned in the projection of the arm line with respect to the acromion (Figure 2D), observing the rotator cuff and the convergence of the supraspinatus and infraspinatus fibers (Figure 2E,F). A video sequence was recorded from the point of measurement identification to the application of glenohumeral joint caudal distraction force (Appendix A). The distance between the most distal point of the acromion and the first visible point of the humeral head at rest (Figure 2E), and distraction (Figure 2F) was drawn at the intersection of the acoustic shadow of the acromion and humeral head [27]. The coracoacromial ligament distance (CAL) (Figure 2D) was also measured with the probe placed longitudinally between the coracoid process (Figure 2D orange dot) and acromion (Figure 2D yellow dot). An axial view of the ligament (Figure 2G) was shown to measure from the last visible point of the acoustic shadow of the coracoid and acromion (Figure 2H) [28]. CAL thickness was also measured at the midpoint of the distance traced from the coracoid and the acromion process [29]. For the bicipital groove angle (BG) assessment (Figure 3A), the probe was placed transversely to identify the most prominent point of bone relief (Figure 3B) [30]. A line was drawn from the most prominent cortical point of the medial edge of the BG to the most concave point inside the groove, leading to the most prominent point of the lateral edge of the groove (Figure 3C) [31]. To identify the biceps brachii longus head tendon (BBLHT) the probe was briefly moved proximally, making probe inclinations to maintain the echogenicity of the entire tendon (Figure 3D), tracing the cross-sectional area (CSA) of the tendon (Figure 3E) [31]. The evaluation of the lower trapezius and serratus anterior was performed by the patient while raising the arm in the scapular plane until 90° of abduction, with a weight of 2.5 kg. An LT scan was performed with a probe located on the lateral edge of the thoracic spinous process, corresponding to the inferior angle of the scapula, oriented transversely to the rib cage (Figure 4A) (Appendix A). The LT thickness was measured at rest (Figure 4B) and during contraction (Figure 4C), tracing a horizontal line of 2 cm from the spinous process, and a vertical line was drawn from the deepest and most superficial aponeurosis of the LT [32]. The serratus anterior (SA) was performed with the probe along the medial axillary edge within the thoracic cage space between the pectoralis major and latissimus dorsi at the inferior border of the scapula (Figure 4D) (Appendix A). The cortical edge of the reference rib at its most crucial point was selected up to the most superficial point of the SA at rest (Figure 4E) and during contraction (Figure 4F). For both the LT and SA assessments, the participant was requested to contract for 3 s, followed by 15 s of rest, to measure the difference between rest and contraction [32].

### 2.6. Statistical Analysis

Statistical analyses were conducted using the Statistical Package for the Social Sciences (SPSS 29.0, IBM, Armonk, NY, USA). The level of significance (α error) was set at 0.05, and statistical significance was determined by a *p*-value < 0.05, with a confidence interval (CI) of 95%. The distribution of variables was evaluated using the Shapiro–Wilk test.

When dealing with normal data (Shapiro–Wilk test with a *p*-value ≥ 0.05), outcomes are presented as mean ± standard deviation (SD) with the range (minimum–maximum). For non-normal distributions, the median ± interquartile range (IQR) with a range (minimum–maximum) was provided. Differences between PS and NPS were examined using Student’s paired t-test for normal data and the Wilcoxon test for non-normally distributed data. To quantify the effect size (ES) of quantitative data, Cohen’s d was calculated, classifying the outcomes as small (from 0.20 to 0.49), medium (from 0.50 to 0.79), or large (>0.8) effect sizes.

## 3. Results

### 3.1. Demographics Data

The analyzed sample comprised twenty male CrossFit athletes (Table 1). The average age of the participants was 25.70 ± 4.03 years, with an average weight of 86.37 ± 8.86 kg and an average height of 1.79 ± 0.07 m. The participants’ BMI was 26.84 ± 2.62 kg/m^2^. The assessment revealed that 75% of the participants were right-arm dominant, of which 66.67% had PS. Although 75% of the athletes were right-arm dominant, no consistent pattern was observed linking shoulder dominance to the presence of pain.

### 3.2. USI Differences

The main result of this study was that it showed a significant difference in SADB thickness of 0.2 mm (*p* < 0.05) thicker for the PS than for NPS with medium ES (d = 0.74). Table 2 summarizes all ultrasound variables with their mean values, standard deviations, and range for both shoulders, including mean differences, 95% confidence intervals, *p*-values, and effect sizes for each comparison. The remaining US measurements did not show any significant differences between shoulders (Table 2). Although this finding reached statistical significance, the observed difference in SADB thickness (0.2 mm) did not exceed the minimum detectable change (MDC = 0.31 mm), which limits its clinical interpretability. A small increase in SST (0.02 cm) was observed in PS, with a small ES (d = 0.34). Difference in AHD_Dif. was 0.02 cm for the PS compared with the NPS, with a small ES (d = 0.21). The CSA of the BBLHT is reduced by 0.01 cm^2^ for PS compared to NPS with a small ES (d = 0.31). The BG angle differed by 0.71° with a small ES (d = 0.08). CAL distance showed a difference of 0.06 cm with a small ES (d = 0.27), and CAL thickness showed a difference of 0.01 mm with a very small ES (d = −0.006). The SA_Dif. for PS showed as 0.06 cm smaller than the NPS, with a small ES (d = 0.22). Finally, the LT_Dif. was 0.01 cm smaller than that of the NPS with a small ES (d = 0.13).

## 4. Discussion

The main purpose of the study was to compare morphology and changes in muscle thickness by ultrasound between the subacromial shoulder pain versus the asymptomatic side of male CrossFit athletes. In summary, our findings showed a statistically significant difference in thickness of the SADB of the painful shoulder compared to the asymptomatic side in CrossFit athletes with SSP. All other ultrasound variables did not show differences between PS and NPS.

The main finding of our study was the presence of an increase in the thickness of the SADB of the PS versus the NPS of 0.2 mm. Repetitive impingement associated with movements performed at extreme high intensity and range of motion is associated with soft tissue microtrauma, leading to thickening of collagen fibers and potential space conflict in the shoulder [4]. Previous research has shown soft-tissue differences in athletes, reporting that ultrasound might play an interesting role as a prognostic tool in athlete prevention programs [33]. To our knowledge, only a few studies have directly evaluated structural alterations in CrossFit athletes using imaging modalities such as MRI or USI. Bernstorff et al. [14] identified frequent bursal and tendinous abnormalities in symptomatic CrossFit athletes, reinforcing the relevance of exploring such findings through more accessible tools like musculoskeletal ultrasound. From a biological perspective, an increased thickness of the SADB in the superior glenohumeral space can be supported with respect to the mechanics of overhead movements. Nonetheless, despite the observed increase in SADB thickness in the PS versus the NPS of 0.20 mm difference, the clinical relevance of these differences cannot be justified based on the reliability study values reported by Hougs Kjaer et al. (MDC = 0.31; SEM = 0.08) [26]. Thus, despite the significant differences (*p* = 0.004) and the moderate effect size reported (d = 0.74), the difference may not reflect a true physiological change beyond measurement error. This highlights the need for cautious interpretation when translating small morphologic changes into clinical decisions. In this context, SADB thickness should not be used in isolation as a diagnostic marker, and its utility in identifying or monitoring subacromial pain must be integrated with functional, symptomatic, and contextual factors.

Several research studies determining a weak relationship between ultrasound findings and shoulder pain have been described [19,20,34,35]. Couanis et al. reported a progressive increase in SADB thickness with greater training duration and volume among open-water marathon swimmers. While this thickening was not associated with pain, it was attributed to an adaptive response to repetitive movement. Pain was only linked to an acute increase in SADB thickness following competition [36]. In this context, the detection of structural abnormalities by imaging tests alone is insufficient to elucidate the relationship between pain and local tissue changes [35]. However, increased SADB thickness has been proposed as a possible structural adaptation in individuals with SSP that may lead to decreased subacromial space. Miyake et al. reported a reduction in CAL thickness in participants with rotator cuff tears [29]. Nevertheless, our results did not report changes between shoulders in the AHD and CAL distance and thickness [28,37,38]. A possible explanation of the lack of structural differences could be associated with the fact that CrossFit is related to bilateral overhead movements. Hence, it could be proposed that structural adaptation could be similar in both shoulders, regardless of the presence of symptoms. Our results align with some previous studies reporting bursal thickening in symptomatic populations [14,19], yet other investigations in overhead athletes have failed to find consistent sonographic differences between painful and asymptomatic shoulders [19,20,35]. These discrepancies highlight the complex, multifactorial nature of shoulder pain and the necessity of contextualizing structural findings within a broader clinical and functional framework.

Otherwise, Siang Ting et al. determined relationships between coracohumeral distance and coracohumeral ligament thickness in patients with SSP [39]. Indeed, the coracohumeral ligament plays an important role in stabilizing the glenohumeral head during inferior displacement and external rotation, extreme movements to which the glenohumeral joint is frequently exposed during the execution of overhead weightlifting gestures often performed in CrossFit.

Impingement of structures during space-closing mechanisms, such as the subacromial space during head movements, can result in mechanical compromise of the tendon. This leads to increased compressive forces on the bursal tissue, paratenon, or bony periosteum, which are richly innervated by free nerve endings and nociceptors, playing an important role in pain and sensitivity. From this perspective, it has been hypothesized that mechanical stress in overhead athletes may contribute to both peripheral and central sensitization mechanisms [40]. CrossFit athletes are exposed to high workloads performed at high intensity, with short rest times and complex sequential gestures that accentuate the mechanical demands placed on the shoulder and may increase the risk of peripheral fatigue and related strength deficits [41]. Although our study was not designed to directly evaluate these neurophysiological processes, the observed structural findings, combined with the intensity of CrossFit training, warrant further investigation of these mechanisms in future research.

### Limitations and Future Research

Some study limitations must be considered, such as the exclusion of women due to the potential risk of bias associated with sex-related differences. Although the sample size was calculated, the findings of this study must be interpreted with caution. In particular, the sample size calculation was based solely on SADB thickness as the primary outcome, without corrections for multiple comparisons, which may limit the statistical power to detect differences in other morphological variables. Moreover, subgroup analyses by arm dominance were not performed, which could have provided additional insights into unilateral adaptations and their potential relation to shoulder pain. Future studies may focus on the assessment of the bilateral adaptive response of sonographic shoulder morphology in CrossFit athletes with SSP, exploring different shoulder spaces such as the postero-internal corner, the rotator interval, or the subcoracoid space, as well as related soft-tissue structures. Additionally, while quantitative sonographic measurements (e.g., thickness, echogenicity) remain essential, the incorporation of structured grading systems based on expert pattern recognition could provide complementary insights into subtle morphological alterations that may not be captured by linear or pixel-based variables alone. Tools such as the Modified Öhberg scale (based on five items for tendon vascularization) [42], Ultrasound Tissue Characterization (UTC) (for fiber organization and echo categorization) [43], or Likert-type expert-based grading systems, along with their intra- and inter-rater validation using defined criteria for identifying tendon abnormalities (e.g., delamination, disruption of fiber pattern, or the presence of echo defects), as proposed by Yoon K et al. [44], may provide a more comprehensive framework for interpreting ultrasound findings in athletes, integrating both structural evaluation and pain-related clinical insight [40,45]. These tools could be operationalized in future studies through consensus-based protocols or training systems for experienced examiners. For example, standardized sonographic assessment protocols, examiner training to ensure inter-session reliability, and the use of structured scoring forms or classification templates may facilitate consistent real-time categorization of sonographic findings. Incorporating these systems into future study designs could enhance the clinical interpretation of shoulder ultrasound specifically in male CrossFit athletes by linking sonographic patterns to pain intensity, perceived function, or return-to-training progression. Building upon this structural perspective, future research should also explore the relationship between structural changes in the SADB and structural components of the shoulder joint, as well as in the evaluation of the posterointernal region of the shoulder and structures such as the posterior glenohumeral capsule, the infraspinatus tendon thickness, and the posterointernal impingement. This could help clarify whether isolated changes in the SADB reflect a localized phenomenon or form part of a broader pattern of tissue adaptation or dysfunction in overhead athletes. Finally, considering the high physical demands and psychological pressures inherent to competitive CrossFit, it would be valuable to assess pain-related beliefs and psychological constructs such as fear avoidance behaviors, kinesiophobia, pain catastrophizing, or self-efficacy in relation to the presence of pain and structural abnormalities. Their integration into future study designs could contribute to a more comprehensive biopsychosocial understanding of shoulder pain in this athletic population.

## 5. Conclusions

No sonographic differences were observed between the PS and NPS in CrossFit athletes. However, small differences in SADB thickness were observed. These findings suggest that the presence of pain in CrossFit athletes cannot be solely attributed to morphological changes, indicating that other factors such as neuromuscular adaptations, movement patterns, or psychosocial contributors may be involved in the development of subacromial shoulder pain in this population. Moreover, considering that the observed sonographic difference did not exceed the minimum detectable change, its clinical significance remains limited. Therefore, the utility of USI as a diagnostic tool for unilateral subacromial shoulder pain in symmetrical sports like CrossFit should be interpreted with caution. Still, given the small sample size and the exclusive inclusion of male Rx/Elite athletes, the generalizability of these results to female or recreational populations remains limited.

## Figures and Tables

**Figure 1 medicina-61-01304-f001:**
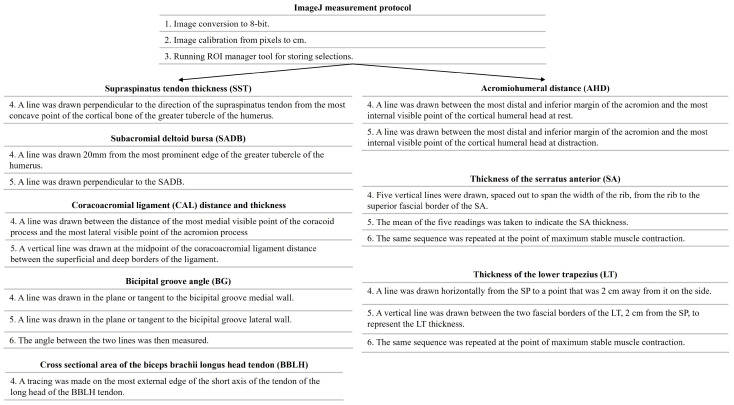
Standardization of the ultrasound image analysis procedure through ImageJ software.

**Figure 2 medicina-61-01304-f002:**
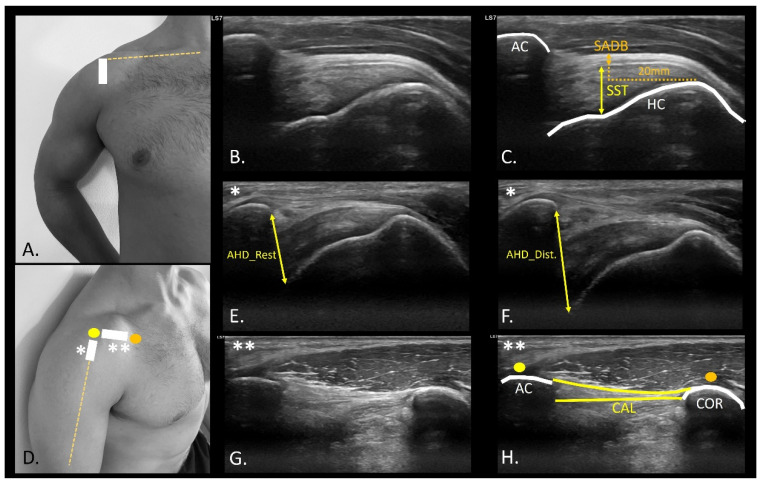
Ultrasound probe location and assessment of the supraspinatus tendon, acromiohumeral distance, and coracoacromial ligament. (**A**), Probe location for longitudinal assessment of supraspinatus tendon (**B**); supraspinatus tendon thickness and SADB thickness measurements (**C**) (Dashed lines, reference distance of 20mm from the greater tubercle of the humerus to take the measurement point of the SST thickness); (**D**), Probe location (*) for the longitudinal assessment of acromiohumeral distance at rest (**E**) and distraction (**F**); Probe location (**) for longitudinal assessment of the coracoacromial ligament (**G**,**H**). Abbreviations: AC, acromion process; AHD_Dist., acromiohumeral distance at distraction; AHD_Rest, acromiohumeral distance at rest; CAL, coracoacromial ligament; COR, coracoid process; HC, humeral head cortical bone; SADB, subacromial deltoid bursa; SST, supraspinatus tendon thickness; TH, thickness.

**Figure 3 medicina-61-01304-f003:**
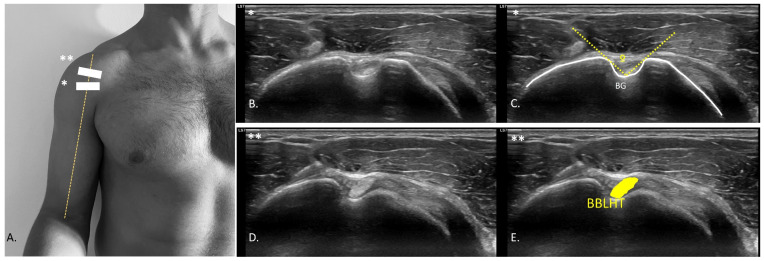
Ultrasound probe location and assessment of the bicipital groove angle and CSA of the biceps brachii longus head tendon. (**A**), Probe location (*) for transverse section assessment of the bicipital groove/intertrochanteric sulcus angle (**B**,**C**) (Dashed lines, showing the angle on the border of the bicipital groove as a measurement reference); (**A**), Probe location (**) transverse section assessment of the CSA of the BBLHT (**D**,**E**). Abbreviations: BBLHT, biceps brachii longus heat tendon; BG, bicipital groove.

**Figure 4 medicina-61-01304-f004:**
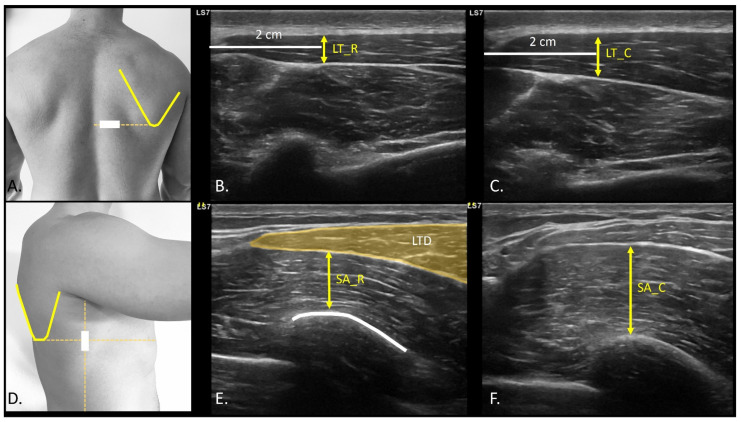
Ultrasound probe location and assessment of the lower trapezius and serratus anterior muscles. (**A**), Probe location for transverse section assessment for lower trapezius at rest (**B**) and contraction (**C**); (**D**), Probe location for serratus anterior assessment at rest (**E**) and contraction (**F**). Abbreviations: LT_Cont., lower trapezius at contraction; LT_Rest, lower trapezius at rest; SA_Cont., serratus anterior at contraction; SA_Rest, serratus anterior at rest.

**Table 1 medicina-61-01304-t001:** Quantitative descriptive variables of male CrossFit athletes.

Descriptive Variables	Total Sample (n = 20)
Age (years)	25.70 ± 4.03 (20.00–32.00)
Weight (kg)	86.37 ± 8.86 (75.00–110.00)
Height (m)	1.79 ± 0.07 (1.70–1.91)
BMI (kg/m^2^)	26.84 ± 2.62 (23.77–34.72)
Dominant Side (R/L) (%)	15/5 (75%/25%)
Painful Side (R/L) (%)	12/8 (66.67%/33.33%)

Abbreviations: BMI, Body Mass Index; L, Left; Min, minimum; Max, maximum; R, Right. Mean ± standard deviation and range (min–max).

**Table 2 medicina-61-01304-t002:** Quantitative data of ultrasound measure of painful and non-painful shoulder in male CrossFit athletes.

USI Variables	PS (n = 20)	NPS (n = 20)	Mean Differences (95%CI)	PS vs. NPS *p*-Value (ES)
SST (cm)	0.62 ± 0.07 (0.45–0.73)	0.60 ± 0.06 (0.44–0.70)	0.02 (−0.01–0.04)	0.136 (0.34)
AHD_Rest (cm)	0.90 ± 0.9 (0.75–1.06)	0.92 ± 0.12 (0.73–1.24)	−0.02 (−0.07–0.02)	0.285 (0.24)
AHD_Dist. (cm)	1.13 ± 0.21 (0.80–1.50)	1.13 ± 0.24 (0.78–1.75)	−0.01 (−0.6–0.05)	0.944 (0.02)
AHD_Dif. (cm)	0.23 ± 0.17 (0.04–0.60)	0.21 ± 0.16 (0.02–0.56)	0.02 (−0.02–0.07)	0.341 (0.21)
BBLHT CSA (cm^2^)	0.11 ± 0.02 (0.08–0.16)	0.12 ± 0.02 (0.08–0.16)	−0.01 (−0.2–0.01)	0.177 (0.31)
BG (°)	118.26 ± 8.17 (95.00–130.80)	117.55 ± 10.66 (95.40–132.00)	0.71 (−3.31–4.74)	0.714 (0.08)
CAL distance (cm)	2.89 ± 0.29 (2.16–3.36)	2.82 ± 0.37 (2.05–3.77)	0.002 (−0.04–0.16)	0.231 (0.27)
CAL thickness (mm)	1.01 ± 0.03 (0.26–1.64)	1.02 ± 0.03 (0.59–1.69)	−0.0002 (−0.018–0.017)	0.977 (0.006)
SADB thickness (mm)	0.9 ± 0.53 (0.4–2.5)	0.7 ± 0.43 (0.2–1.7)	0.2 (0.07–0.32)	**0.004 (0.74) ***
SA_Rest (cm)	0.75 ± 0.22 (0.40–1.26)	0.77 ± 0.30 (0.26–1.58)	−0.01 (−0.12–0.09)	0.758 (0.07)
SA_Cont. (cm)	1.10 ± 0.30 (0.64–1.59)	1.18 ± 0.39 (0.46–2.05)	−0.08 (−0.24–0.08)	0.326 (0.23)
SA_Dif. (cm)	0.35 ± 0.16 (0.07–0.78)	0.41 ± 0.26 (0.02–0.99)	−0.06 (−0.19–0.07)	0.338 (0.22)
LT_Rest (cm)	0.46 ± 0.12 (0.25–0.72)	0.49 ± 0.21 (0.20–1.21)	−0.03 (−0.11–0.05)	0.477 (0.16)
LT_Cont. (cm)	0.62 ± 0.17 (0.30–0.94)	0.66 ± 0.28 (0.24–1.52)	−0.04 (−0.16–0.07)	0.450 (0.17)
LT_Dif. (cm)	0.15 ± 0.13 (0.01–0.57)	0.17 ± 0.14 (0.02–0.67)	−0.01 (−0.06–0.03)	0.543 (0.13)

Abbreviations: AHD_Dif., acromiohumeral distance difference between rest and distraction; AHD_Dist., acromiohumeral distance at distraction; AHD_Rest, acromiohumeral distance at rest; BBLHT CSA, cross-sectional area of the biceps brachii longus head tendon; BG, bicipital groove angle; CAL; coracoacromial ligament; ES, effect size; LT_Cont., lower trapezius thickness at contraction; LT_Dif., lower trapezius thickness difference between rest and contraction; LT_Rest, lower trapezius thickness at rest; NPS, non-painful shoulder; PS, painful shoulder; SADB, subacromial deltoidea bursa; SA_Cont., serratus anterior thickness at contraction; SA_Dif., serratus anterior thickness difference between rest and contraction; SA_Rest, serratus anterior thickness at rest; SST, supraspinatus tendon thickness. * Student’s *t*-test for paired samples was used according to the parametric distributions. For all analyses, *p* < 0.05 (for a confidence interval of 95%) was considered statistically significant (**bold**).

## Data Availability

Due to ethical and legal considerations, the data supporting the findings of this study are not publicly available. However, they may be made available by the corresponding author upon reasonable request.

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
