# Peer review of "Ultrasound-Based Morphological and Functional Assessment in Male CrossFit Athletes with Unilateral Subacromial Shoulder Pain: An Observational Study"

_medicina, 2025, doi:10.3390/medicina61071304_

Round 1

Reviewer 1 Report

Comments and Suggestions for Authors

Dear Authors,

This is a well-structured and methodologically sound observational study that explores the sonographic differences between symptomatic and asymptomatic shoulders in male CrossFit athletes. The focus on subacromial subdeltoid bursa (SADB) thickness is relevant and timely, considering the increasing popularity of CrossFit and the prevalence of shoulder pain in this population. However, I believe the following revisions and clarifications would significantly strengthen the manuscript:

  1. Participant Training Background:
    While all participants met the inclusion criteria of at least two years of CrossFit experience, more detailed information regarding their training level, competitive status, or specific exercise routines would improve the interpretability of the results. Given the wide variability in CrossFit training intensity and technique, such heterogeneity could influence shoulder morphology and pain perception.
  2. Sample Size and Generalizability:
    Although the sample size calculation is adequately described, the final sample of 20 participants limits the external validity of the findings. The authors may wish to acknowledge more explicitly that the study’s conclusions may not generalize to broader populations, including female athletes or recreational-level practitioners.
  3. Clinical Relevance of Findings:
    Although SADB thickness showed a statistically significant difference between symptomatic and asymptomatic shoulders, the mean difference (0.2 mm) was below the minimum detectable change (MDC = 0.31 mm). This limitation is noted, but further elaboration on its implications for clinical utility and diagnostic decision-making would be beneficial.
  4. Standardization of Terminology:
    Consider standardizing terms such as “painful shoulder (PS)” and “non-painful shoulder (NPS)” across tables and narrative text to avoid confusion.
  5. Future Research Suggestions:
    The discussion briefly mentions potential use of imaging grading scales and bilateral adaptations in CrossFit athletes. Expanding on how such tools might be operationalized in future studies would enrich the conclusion and highlight the study’s contribution to clinical practice.  

       Sincerely,

Author Response

Responses to Reviewers’s comments:

Response to Reviewer 1 comment:

In response to the general comment “This is a well-structured and methodologically sound observational study that explores the sonographic differences between symptomatic and asymptomatic shoulders in male CrossFit athletes. The focus on subacromial subdeltoid bursa (SADB) thickness is relevant and timely, considering the increasing popularity of CrossFit and the prevalence of shoulder pain in this population. However, I believe the following revisions and clarifications would significantly strengthen the manuscript:”

  • We would like to sincerely thank the reviewer for their constructive and insightful comments, which have helped us to improve the clarity, methodological rigor, and overall quality of our manuscript. We greatly appreciate the positive evaluation of our study’s structure, design, and relevance—particularly the recognition of the novelty and timeliness of our focus on subacromial subdeltoid bursa (SADB) thickness in male CrossFit athletes. Below, we address each of the reviewer’s suggestions point by point, highlighting the changes made in the revised version of the manuscript.
  • Comments and Suggestions for Authors

In response to “1. Participant Training Background: While all participants met the inclusion criteria of at least two years of CrossFit experience, more detailed information regarding their training level, competitive status, or specific exercise routines would improve the interpretability of the results. Given the wide variability in CrossFit training intensity and technique, such heterogeneity could influence shoulder morphology and pain perception. ”

  • Thank you for your comment and appreciation. We have incorporated the following information into the participants section, adding more specific information on the characteristics of the population.
    • 3. Participants: “Twenty male (N=20) CrossFit athletes with unilateral shoulder pain were recruited, evaluated bilaterally (n=40) and divided into two groups of 20 PS (n=20) and 20 NPS (n=20) groups. All participants were informed and gave signed consent for participation in the study. Data were collected between January and June 2023 and inclusion criteria for study enrollment were: (a) male CrossFit athletes; (b) aged 18-35; (c) with at least two years of experience in CrossFit discipline; (d) athletes who follow (e) competitive athletes classified in the Rx or Elite category, according to CrossFit’s standardized performance classification system; (f) athletes following a structured and individualized training program, and (g) presenting at least two of the following five shoulder impingement criteria: positive Neer sign, positive Hawkins’s sign, positive Jobe sign, pain with apprehension 18. Additionally, (h) participants had to present pain in overhead weightlifting activities in the last 3 months, such as snatch and jerk variants during training or competition. Participants were excluded if they reported a history of surgical interventions or congenital alterations; metabolic, neurological, autoimmune, or cardiovascular diseases affecting the shoulder; inability to understand the study protocol; cognitive impairments; taking analgesic or anti-inflammatory medication during the study or a week before; or a low physical activity level on the International Physical Activity Questionnaire (IPAQ) in the last week.“

In response to “2. Sample Size and Generalizability: Although the sample size calculation is adequately described, the final sample of 20 participants limits the external validity of the findings. The authors may wish to acknowledge more explicitly that the study’s conclusions may not generalize to broader populations, including female athletes or recreational-level practitioners.”

  • Thank you for your comment and appreciation. We've added a sentence to the conclusions section based on your recommendation.
    • Conclusion: No sonographic differences were observed between the PS and NPS in CrossFit athletes. However, small differences in SADB thickness were observed. These findings suggest that the presence of pain in CrossFit athletes cannot be solely attributed to morphological changes, indicating that other factors such as neuromuscular adaptations, movement patterns, or psychosocial contributors may be involved in the development of subacromial shoulder pain in this population. Moreover, considering that the observed sonographic difference did not exceed the minimum detectable change, its clinical significance remains limited. Therefore, the utility of USI as a diagnostic tool for unilateral subacromial shoulder pain in symmetrical sports like CrossFit should be interpreted with caution. Still, given the small sample size and the exclusive inclusion of male Rx/Elite athletes, the generalizability of these results to female or recreational populations remains limited.

In response to “3. Clinical Relevance of Findings: Although SADB thickness showed a statistically significant difference between symptomatic and asymptomatic shoulders, the mean difference (0.2 mm) was below the minimum detectable change (MDC = 0.31 mm). This limitation is noted, but further elaboration on its implications for clinical utility and diagnostic decision-making would be beneficial.”

  • Thank you for your comment and appreciation. We have incorporated the following information into the discussion section in an attempt to further elaborate on the clinical implications of the observed change in bursa thickness, considering the minimal detectable change reported in previous studies.
    • “4. Discussion: The main finding of our study was the presence of an increase in the thickness of the SADB of the PS versus the NPS of 0.2 mm. Repetitive impingement associated with movements performed at extreme high intensity and range of motion is associated with soft tissue microtrauma, leading to thickening of collagen fibers and potential space conflict in the shoulder 27. Previous research has shown soft-tissue differences in athletes, reporting that ultrasound might play an interesting role as a prognostic tool in athlete prevention programs 28. From a biological perspective, an increased thickness of the SADB in the superior glenohumeral space can be supported with respect to the mechanics of overhead movements. Nonetheless, despite the observed increase in SADB thickness in the PS versus the NPS of 0.20 mm difference, the clinical relevance of these differences cannot be justified based on the reliability study values reported by Kjaer et al. (MDC = 0.31; SEM = 0.08) 17. Thus, although the significant differences (p = 0.004) and the moderate effect size reported (d = 0.74), the difference may not reflect a true physiological change beyond measurement error. This highlights the need for cautious interpretation when translating small morphologic changes into clinical decisions. In this context, SADB thickness should not be used in isolation as a diagnostic marker, and its utility in identifying or monitoring subacromial pain must be integrated with functional, symptomatic, and contextual factors.”

In response to “4. Standardization of Terminology: Consider standardizing terms such as “painful shoulder (PS)” and “non-painful shoulder (NPS)” across tables and narrative text to avoid confusion.”

  • Thank you for your feedback. We have incorporated the abbreviations for PS and NPS into the abbreviations in Table 2 for ease of understanding. We have revised the rest of the text so that PS and NPS abbreviations appear appropriately in the text.

In response to “5. Future Research Suggestions: The discussion briefly mentions potential use of imaging grading scales and bilateral adaptations in CrossFit athletes. Expanding on how such tools might be operationalized in future studies would enrich the conclusion and highlight the study’s contribution to clinical practice.”

  • Thank you for your feedback and for the opportunity to improve the quality of the manuscript. We have made some adjustments to section 4.1 Limitations and Future Research in an attempt to improve the understanding and practical application of the proposed future study designs in this line of research. We have also incorporated some references into the text.
    • “4.1. Limitations and future research: Some study limitations must be considered, such as the exclusion of women due to the potential risk of bias associated with sex-related differences. Although the sample size was calculated, the findings of this study must be interpreted with caution. Moreover, subgroup analyses by arm dominance were not performed, which could have provided additional insights into unilateral adaptations and their potential relation to shoulder pain. Future studies may focus on the assessment of the bilateral adaptive response of sonographic shoulder morphology in CrossFit athletes with SSP, exploring different shoulder spaces such as the postero-internal corner, the rotator interval, or the subcoracoid space, as well as related soft-tissue structures. Additionally, while quantitative sonographic measurements (e.g., thickness, echogenicity) remain essential, the incorporation of structured grading systems based on expert pattern recognition could provide complementary insights into subtle morphological alterations that may not be captured by linear or pixel-based variables alone. Tools such as the Modified Öhberg scale (based on five items for tendon vascularization) 37, Ultrasound Tissue Characterization (UTC) (for fiber organization and echo categorization) 38, or Likert-type expert-based grading systems, along with their intra- and inter-rater validation using defined criteria for identifying tendon abnormalities (e.g., delamination, disruption of fiber pattern, or the presence of echo defects), as proposed by Yoon K et al.39, may provide a more comprehensive framework for interpreting ultrasound findings in athletes, integrating both structural evaluation and pain-related clinical insight 35,40. These tools could be operationalized in future studies through consensus-based protocols or training systems for experienced examiners. For example, standardized sonographic assessment protocols, examiner training to ensure inter-session reliability, and the use of structured scoring forms or classification templates may facilitate consistent real-time categorization of sonographic findings. Incorporating these systems into future study designs could enhance the clinical interpretation of shoulder ultrasound specifically in male CrossFit athletes by linking sonographic patterns to pain intensity, perceived function, or return-to-training progression. Building upon this structural perspective, future research should also explore the relationship between structural changes in the SADB and structural components of the shoulder joint, as well as in the evaluation of the posterointernal region of the shoulder and structures such as the posterior glenohumeral capsule, the infraspinatus tendon thickness and the posterointernal impingement. This could help clarify whether isolated changes in the SADB reflect a localized phenomenon or form part of a broader pattern of tissue adaptation or dysfunction in overhead athletes. Finally, considering the high physical demands and psychological pressures inherent to competitive CrossFit, it would be valuable to assess pain-related beliefs and psychological constructs such as fear avoidance behaviors, kinesiophobia, pain catastrophizing, or self-efficacy in relation to the presence of pain and structural abnormalities. Their integration into future study designs could contribute to a more comprehensive biopsychosocial understanding of shoulder pain in this athletic population.”
  1. van der Vlist AC, Veen JM, van Oosterom RF, van Veldhoven PLJ, Verhaar JAN, de Vos RJ. Ultrasound Doppler Flow in Patients With Chronic Midportion Achilles Tendinopathy: Is Surface Area Quantification a Reliable Method? J Ultrasound Med [Internet]. 2020 Apr 1 [cited 2024 Jan 26];39(4):731–9. Available from: https://pubmed.ncbi.nlm.nih.gov/31724758/
  2. Noble JA. Ultrasound image segmentation and tissue characterization. Proc Inst Mech Eng H. 2010;224(2):307–16.
  3. Yoon K, Kim H, Han S Bin, Song HS. Ultrasound Findings Aid Decisions to Repair Partial Articular Supraspinatus Tendon Avulsion. Journal of Ultrasound in Medicine. 2020 Oct 1;39(10):2005–11.

Thanks for your valuable commentaries which have permitted us to improve the quality of the manuscript.

Sincerely,

The authors

Reviewer 2 Report

Comments and Suggestions for Authors

This cross-sectional observational study investigates whether there are structural and functional differences between symptomatic and asymptomatic shoulders in male CrossFit athletes with unilateral subacromial shoulder pain (SSP), using ultrasound imaging (USI). The key finding is a statistically significant increase in subacromial-subdeltoid bursa (SADB) thickness in the painful shoulders, while no significant differences were observed in other ultrasound parameters.

Title and Abstract: The title is clear and accurately reflects the study's content. The abstract provides a concise summary of the background, methods, main findings, and conclusion. However, the abstract mentions a range of US parameters but does not clearly state that only one (SADB thickness) showed a statistically significant difference.

Introduction: Good rationale for studying SSP in CrossFit athletes; clearly states the research gap, but lacks a strong theoretical framework linking USI findings with the pathomechanics of SSP. Several statements are supported with weakly integrated references. Some grammatical errors and awkward phrasings (e.g., "redisposed" instead of "predisposed") – please perform proof reading in more detail.

Methods: Clearly described procedures; adherence to STROBE guidelines; ethical approval obtained. The blinding protocol of the examiner could be more thoroughly described. No discussion of intra- or inter-rater reliability of USI in this context (although referenced).

Results: Data are clearly presented; proper statistical tests used; effect sizes reported. The only statistically significant finding (SADB thickness) has questionable clinical relevance (does not exceed MDC). No subgroup analysis (e.g., dominant vs. non-dominant side). Table 2 is comprehensive but lacks visual clarity due to overload of variables.

Discussion: Acknowledges the main limitation (lack of clinical relevance despite statistical significance),  but still overreaches by speculating about central sensitization and neural mechanisms without supporting data from this study. Some key references are outdated or not directly related to CrossFit athletes. The discussion lacks a critical comparison with studies that found no USI differences in overhead athletes.

Conclusion: Accurately reflects the results, but understates the clinical insignificance of the USI difference. Lacks emphasis on the potential lack of utility of USI for unilateral SSP diagnosis in symmetrical sports like CrossFit.

General statement
While the study addresses a relevant and underexplored topic, its contribution is limited by the marginal clinical significance of its sole positive finding, an overly descriptive discussion, and several methodological flaws.

A minor revision should improve the clarity of rationale, acknowledge limitations more transparently, refine speculative discussion, and better contextualize findings within the broader literature.

Author Response

Responses to Reviewers’s comments:

Response to Reviewer 2 comment:

In response to: “This cross-sectional observational study investigates whether there are structural and functional differences between symptomatic and asymptomatic shoulders in male CrossFit athletes with unilateral subacromial shoulder pain (SSP), using ultrasound imaging (USI). The key finding is a statistically significant increase in subacromial-subdeltoid bursa (SADB) thickness in the painful shoulders, while no significant differences were observed in other ultrasound parameters; General statement: While the study addresses a relevant and underexplored topic, its contribution is limited by the marginal clinical significance of its sole positive finding, an overly descriptive discussion, and several methodological flaws.

A minor revision should improve the clarity of rationale, acknowledge limitations more transparently, refine speculative discussion, and better contextualize findings within the broader literature.”

  • We would like to express our sincere gratitude for the time and effort you devoted to reviewing our manuscript, as well as for your thoughtful and constructive comments. We greatly appreciate your recognition of the relevance of the topic addressed, and we have carefully considered the methodological and interpretative aspects you identified as requiring improvement.

In response to your suggestions, we have revised the manuscript to enhance the clarity of the study rationale, acknowledge the study’s limitations more transparently, and improve the contextualization of our findings within the broader scientific literature. Furthermore, we have refined speculative elements in the discussion to ensure a more balanced and clinically grounded interpretation of the results. We trust that these modifications have contributed to strengthening the overall quality, coherence, and impact of the manuscript.

In response to “Title and Abstract: The title is clear and accurately reflects the study's content. The abstract provides a concise summary of the background, methods, main findings, and conclusion. However, the abstract mentions a range of US parameters but does not clearly state that only one (SADB thickness) showed a statistically significant difference.”

  • We thank you for your valuable observation regarding the clarity of the abstract. As you correctly noted, although the abstract previously listed all ultrasound parameters assessed, it did not explicitly state that only one variable (SADB thickness) showed a statistically significant difference. To address this, we have revised the Results and Conclusion sections of the abstract to clearly indicate that only SADB thickness differed significantly between the painful and non-painful shoulders, while all other parameters showed no significant differences.
    Additionally, the revised conclusion clarifies that this isolated morphological finding may not fully explain the presence of shoulder pain, which cannot be solely attributed to structural changes. We believe these modifications improve the precision and transparency of the abstract in line with your suggestion.
    • “Abstract: Background: CrossFit is a discipline involving a wide range of overhead movements performed at high intensity and under accumulated fatigue that predispose to a high risk of shoulder complex injuries. Objectives: This study aimed to compare ultrasonographic findings between symptomatic and asymptomatic shoulders in CrossFit athletes. Methods: A cross-sectional study was conducted to compare ultrasound parameters between the painful and non-painful shoulders in CrossFit athletes with unilateral subacromial shoulder pain. Assessed variables included subacromial subdeltoid bursa thickness, supraspinatus tendon thickness, the acromiohumeral distance, the coracoacromial ligament distance, the bicipital groove angle, cross-sectional area of the biceps brachii longus head tendon, as well as the serratus anterior and lower trapezius muscle thickness. Results: Twenty male CrossFit athletes (forty shoulders) with an average age of 25.70 ± 4.03 years participated in the study. A statistically significant increase was observed (p < 0.05) in the subacromial subdeltoid bursa thickness in the painful shoulder compared to the asymptomatic side. All other ultrasound parameters did not show statistically significant differences. Conclusion: Only subacromial subdeltoid bursa thickness differed significantly between sides. This isolated finding may not fully explain shoulder pain, which cannot be solely attributed to morphological changes. Further research is needed to determine the relationship between shoulder pain and ultrasound features in CrossFit athletes, as well as the role of ultrasound in predicting structural changes in pain conditions.”

In response to “Introduction: Good rationale for studying SSP in CrossFit athletes; clearly states the research gap, but lacks a strong theoretical framework linking USI findings with the pathomechanics of SSP. Several statements are supported with weakly integrated references. Some grammatical errors and awkward phrasings (e.g., "redisposed" instead of "predisposed") – please perform proof reading in more detail.”

  • Thank you for your valuable feedback. We have substantially revised the Introduction to better articulate the theoretical framework linking ultrasound imaging (USI) findings with the pathomechanics of subacromial shoulder pain (SSP) in CrossFit athletes. Several new references have been added to strengthen the scientific background and justify the use of USI as a practical and accessible imaging tool. In addition, we have corrected grammatical issues and improved the clarity and integration of key concepts throughout the section.
    • Introduction: CrossFit is a sports discipline, based on the combination of strength and endurance exercises. It involves constantly varied functional movements, applied intermittently at high intensity, with special involvement of overhead kinetic chains 1. Thus, CrossFit athletes are particularly predisposed to shoulder injuries, with a 28.6% prevalence rate, especially in lifting activities or gymnastics tasks 2. Shoulder injuries are 2.79 higher in men than in women 3. Overhead athletes are predisposed to develop subacromial shoulder pain (SSP), a condition frequently linked to mechanical overload of subacromial structures such as the supraspinatus tendon, the subacromial-subdeltoid bursa, or the coracoacromial arch. In this context, the biomechanics of repetitive overhead movements play a key role in both the onset and persistence of pain, and should therefore guide the clinical and imaging evaluation of the painful athletic shoulder 4–6. Despite the difficulties in classifying shoulder pain from a pathomechanical and/or a functional point of view, SSP as the term "umbrella" encompasses shoulder pain associated with glenohumeral structures typically described in overhead movements, such as the snatch or the jerk variants in CrossFit 7–9. In this context, the need for a wide range of pain-free movements, combined with optimal motor control and strength, makes CrossFit a highly complex and demanding movement pattern. The considerable demand for push and pull movement patterns of the upper extremity makes this discipline a functionally demanding, requiring a comprehensive evaluation that integrates biomechanical, functional, and sensorimotor components 10,11. Considering the shoulder complex assessment, an integrated approach combining physical, functional, and psychological variables plays a fundamental role in the clinical examination of overhead athletes 12. Recently, the development of ultrasound imaging (USI) and its accessibility poses a challenge when it comes to combining the information obtained from an imaging tool with the information obtained through the examination 13. USI is considered a safe, non-invasive, and cost-effective tool for musculoskeletal shoulder evaluation, especially when compared to more expensive or static imaging modalities such as MRI. While MRI has been used to characterize shoulder pathologies in CrossFit athletes 14, USI provides a more accessible alternative for routine structural and functional assessment. Its dynamic capability enables the clinician to assess structural integrity and movement-related alterations in real time, which is particularly relevant in pathologies with a functional component, such as SSP. In addition, it facilitates the identification of anomalies in symptomatic shoulders with SSP, including alterations in supraspinatus tendon thickness (SST), subacromial deltoid bursa (SADB), and acromiohumeral distance (AHD). These alterations may reflect adaptations to repetitive mechanical loading, bursal irritation, or reduced subacromial space, all of which are biomechanically plausible in high-volume overhead athletes such as CrossFit practitioners 15,16. Moreover, USI allows for the dynamic evaluation of muscle activation patterns. Changes in muscle thickness between rest and contraction have been used to examine functional behavior of stabilizing muscles such as the lower trapezius (LT) or serratus anterior (SA), which may be compromised in athletes with shoulder pain. This functional dimension enhances the diagnostic value of USI, allowing clinicians to assess both static morphology and contractile behavior under load 17,18. Several studies have attempted to identify structural differences between individuals with and without shoulder pain, as well as between the painful shoulder (PS) and the non-painful shoulder (NPS) within the same individual 19,20. However, to our knowledge, no previous study has conducted a bilateral comparison of both structural and functional ultrasound findings between symptomatic and asymptomatic shoulders specifically in CrossFit athletes. This is particularly relevant given the sport's high exposure to complex overhead gestures and the documented prevalence of SSP. Thus, this study aimed to highlight the structural and functional differences identified by USI between the PS and NPS in male CrossFit athletes with unilateral SSP We hypothesized that the painful shoulder would present measurable structural and functional differences compared with the asymptomatic shoulder.”

Cools AM, Michener LA. Shoulder pain: Can one label satisfy everyone and everything? Br J Sports Med [Internet]. 2017 Mar 1 [cited 2025 Jul 14];51(5):416–7. Available from: https://pubmed.ncbi.nlm.nih.gov/27806952/

Lewis J. Rotator cuff related shoulder pain: Assessment, management and uncertainties. Man Ther. 2016 Jun;23:57–68.

Warth RJ, Millett PJ. Physical examination of the shoulder: An evidence-based approach. Physical Examination of the Shoulder: An Evidence-Based Approach. 2015 Jan 1;1–276.

Bernstorff MA, Schumann N, Schwake L, Somberg O, Balke M, Schildhauer T, et al. Shoulder pathologies in CrossFit: a magnetic resonance imaging study of 51 cases. Journal of Sports Medicine and Physical Fitness [Internet]. 2024 May 1 [cited 2025 Jul 14];64(5):475–82. Available from: https://pubmed.ncbi.nlm.nih.gov/38445843/

Purim KSM, Zilli A, de Almeida Leite GF, Susuki GK, Hapner LMS, Zangari MAC. Musculoskeletal Injuries in Competitive CrossFit Athletes. Rev Bras Ortop (Sao Paulo) [Internet]. 2024 Dec 21 [cited 2025 Jul 14];59(6):e976–80. Available from: https://pubmed.ncbi.nlm.nih.gov/39711647/

Lastra-Rodríguez L, Llamas-Ramos I, Rodríguez-Pérez V, Llamas-Ramos R, López-Rodríguez AF. Musculoskeletal Injuries and Risk Factors in Spanish CrossFit® Practitioners. Healthcare [Internet]. 2023 May 1 [cited 2025 Jul 14];11(9):1346. Available from: https://pmc.ncbi.nlm.nih.gov/articles/PMC10178070/

In response to “Methods: Clearly described procedures; adherence to STROBE guidelines; ethical approval obtained. The blinding protocol of the examiner could be more thoroughly described. No discussion of intra- or inter-rater reliability of USI in this context (although referenced).”

  • We appreciate your comment. We have introduced modifications aimed at improving the clarity and completeness of the Methods section. In particular, the blinding procedure has been clarified, specifying that all assessments followed a fixed sequence (right before left side) to minimize bias. Additionally, we acknowledge the absence of intra- and inter-rater reliability testing in the present study, as noted.
    • “2.5. USI assessment: The USI assessment was conducted by an experienced evaluator with over five years of expertise in musculoskeletal USI assessment, who was blinded of the PS of each participant. To minimize potential examiner bias, participants were instructed not to disclose their symptomatic side during evaluation. All assessments were systematically performed in the same sequence, beginning with the right shoulder followed by the left shoulder, regardless of symptom presentation. All participants were evaluated in a sitting position, using ultrasound equipment (Logiq S7 Expert US, GE Healthcare, Chicago, IL, USA) with a wide-spectrum linear probe (ML6-15 H40452LY Wide-Spectrum Linear Array Probe, field of view 50 mm) with a frequency range of 4–15 MHz, and a preset system was established to standardize USI evaluation (depth, 4.5 cm, frequency, 12 MHz; fifty-five gain points; sixty-nine dynamic range points; and one focus positioned 2 cm deep). All images and videos were stored in DICOM format and analyzed using the open-source software ImageJ Fiji (U.S. National Institutes of Health; Bethesda, Maryland, USA) (Figure 1.) 23. All scans and measurements were conducted three times to obtain the average of the three measurements. Although all measurements were performed by a single experienced examiner to ensure consistency, intra- and inter-rater reliability were not assessed within this study. However, the selected measurement protocols have demonstrated excellent intra-rater reliability in prior research, particularly for SADB and SST variables 23.”

In response to “Results: Data are clearly presented; proper statistical tests used; effect sizes reported. The only statistically significant finding (SADB thickness) has questionable clinical relevance (does not exceed MDC). No subgroup analysis (e.g., dominant vs. non-dominant side). Table 2 is comprehensive but lacks visual clarity due to overload of variables.”

  • Thank you for your valuable comment and review of the results section. We have incorporated a concise explanatory sentence to improve the clarity and comprehension of Table 2 without altering its format. Additionally, we have included a note on the limited clinical relevance of the only statistically significant finding and addressed the lack of subgroup analyses in the revised limitations section.
    • 1. Demographics data: The analyzed sample comprised of twenty male CrossFit athletes (Table 1). The average age of the participants was 25.70 ± 4.03 years, with an average weight of 86.37 ± 8.86 kg and an average height of 1.79 ± 0.07 m. The participants' BMI was 26.84 ± 2.62 kg/m2. The assessment revealed that 75% of the participants were right-arm dominant, of which 66.67% had PS. Although 75% of the athletes were right-arm dominant, no consistent pattern was observed linking shoulder dominance to the presence of pain.
    • 2. USI differences: The main result of this study was that it showed a significant difference in SADB thickness of 0.2 mm (p < 0.05) thicker for the PS than for NPS with medium ES (d = 0.74). Table 2 summarizes all ultrasound variables with their mean values, standard deviations, and range for both shoulders, including mean differences, 95% confidence intervals, p-values, and effect sizes for each comparison. The remaining US measurements did not show any significant differences between shoulders (Table 2). Although this finding reached statistical significance, the observed difference in SADB thickness (0.2 mm) did not exceed the minimum detectable change (MDC = 0.31 mm), which limits its clinical interpretability. A small increase in SST (0.02 cm) was observed in PS, with a small ES (d = 0.34). Difference in AHD_Dif. was 0.02 cm for the PS compared with the NPS, with a small ES (d = 0.21). The CSA of the BBLHT is reduced by 0.01 cm2 for PS compared to NPS with a small ES (d = 0.31). The BG angle differed by 0.71 ° with a small ES (d = 0.08). CAL distance showed a difference of 0.06 cm with a small ES (d = 0.27), and CAL thickness showed a difference of 0.01 mm with a very small ES (d = -0.006). The SA_Dif. for PS showed an 0.06 cm smaller than the NPS, with a small ES (d = 0.22). Finally, the LT_Dif. was 0.01 cm smaller than that of the NPS with a small ES (d = 0.13).”
    • “4.1. Limitations and future research: Some study limitations must be considered, such as the exclusion of women due to the potential risk of bias associated with sex-related differences. Although the sample size was calculated, the findings of this study must be interpreted with caution. Moreover, subgroup analyses by arm dominance were not performed, which could have provided additional insights into unilateral adaptations and their potential relation to shoulder pain.
    • “3.2. USI differences: The main result of this study was that it showed a significant difference in SADB thickness of 0.2 mm (p < 0.05) thicker for the PS than for NPS with medium ES (d = 0.74) (Table 2). The remaining US measurements did not show any significant differences between shoulders (Table 2). Although this finding reached statistical significance, the observed difference in SADB thickness (0.2 mm) did not exceed the minimum detectable change (MDC = 0.31 mm), which limits its clinical interpretability. A small increase in SST (0.02 cm) was observed in PS, with a small ES (d = 0.34). Difference in AHD_Dif. was 0.02 cm for the PS compared with the NPS, with a small ES (d = 0.21). The CSA of the BBLHT is reduced by 0.01 cm2 for PS compared to NPS with a small ES (d = 0.31). The BG angle differed by 0.71 ° with a small ES (d = 0.08). CAL distance showed a difference of 0.06 cm with a small ES (d = 0.27), and CAL thickness showed a difference of 0.01 mm with a very small ES (d = -0.006). The SA_Dif. for PS showed an 0.06 cm smaller than the NPS, with a small ES (d = 0.22). Finally, the LT_Dif. was 0.01 cm smaller than that of the NPS with a small ES (d = 0.13).

In response to “Discussion: Acknowledges the main limitation (lack of clinical relevance despite statistical significance), but still overreaches by speculating about central sensitization and neural mechanisms without supporting data from this study. Some key references are outdated or not directly related to CrossFit athletes. The discussion lacks a critical comparison with studies that found no USI differences in overhead athletes.”

  • Thank you for your valuable observations and the opportunity to improve the quality of the Discussion. We have revised this section to adopt a more cautious tone regarding speculative interpretations, updated key references, and incorporated a more balanced comparison with studies reporting conflicting findings, as per your suggestions.
    • “4. Discussion: The main finding of our study was the presence of an increase in the thickness of the SADB of the PS versus the NPS of 0.2 mm. Repetitive impingement associated with movements performed at extreme high intensity and range of motion is associated with soft tissue microtrauma, leading to thickening of collagen fibers and potential space conflict in the shoulder 32. Previous research has shown soft-tissue differences in athletes, reporting that ultrasound might play an interesting role as a prognostic tool in athlete prevention programs 33. To our knowledge, only a few studies have directly evaluated structural alterations in CrossFit athletes using imaging modalities such as MRI or USI. Bernstorff et al. 14 identified frequent bursal and tendinous abnormalities in symptomatic CrossFit athletes, reinforcing the relevance of exploring such findings through more accessible tools like musculoskeletal ultrasound.”
    • “4. Discussion: Several research determined a weak relationship between ultrasound findings and shoulder pain has been described 19,20,34,35. Couanis et al. reported a progressive increase in SADB thickness with greater training duration and volume among open-water marathon swimmers. While this thickening was not associated with pain, it was attributed to an adaptive response to repetitive movement. Pain was only linked to an acute increase in SADB thickness following competition 36. In this context, the detection of structural abnormalities by imaging tests alone is insufficient to elucidate the relationship between pain and local tissue changes 35. However, increased SADB thickness has been proposed as a possible structural adaptation in individuals with SSP that may lead to decreased subacromial space. Miyake et al. reported a reduction in CAL thickness in participants with rotator cuff tears 28. Nevertheless, our results did not report changes between shoulders in the AHD and CAL distance and thickness 27,37,38. A possible explanation of the lack of structural differences could be associated with the fact that CrossFit is related to bilateral overhead movements. Hence, it could be proposed that structural adaptation could be similar in both shoulders, regardless of the presence of symptoms. Our results align with some previous studies reporting bursal thickening in symptomatic populations 14,19, yet other investigations in overhead athletes have failed to find consistent sonographic differences between painful and asymptomatic shoulders 19,20,35. These discrepancies highlight the complex, multifactorial nature of shoulder pain and the necessity of contextualizing structural findings within a broader clinical and functional framework.
    • “4. Discussion: Impingement of structures during space-closing mechanisms, such as the subacromial space during head movements, can result in mechanical compromise of the tendon. This leads to increased compressive forces on the bursal tissue, paratenon, or bony periosteum, which are richly innervated by free nerve endings and nociceptors, playing an important role in pain and sensitivity. From this perspective, it has been hypothesized that mechanical stress in overhead athletes may contribute to both peripheral and central sensitization mechanisms 40. CrossFit athletes are exposed to high workloads performed at high intensity, with short rest times and complex sequential gestures that accentuate the mechanical demands placed on the shoulder and may increase the risk of peripheral fatigue and related strength deficits 41. Although our study was not designed to directly evaluate these neurophysiological processes, the observed structural findings, combined with the intensity of CrossFit training, warrant further investigation of these mechanisms in future research.”

In response to “Conclusion: Accurately reflects the results, but understates the clinical insignificance of the USI difference. Lacks emphasis on the potential lack of utility of USI for unilateral SSP diagnosis in symmetrical sports like CrossFit.”

  • We appreciate your feedback. We have adapted the manuscript's conclusion to reflect your considerations, as reflected in your comment.
    • Conclusion: No sonographic differences were observed between the PS and NPS in CrossFit athletes. However, small differences in SADB thickness were observed. These findings suggest that the presence of pain in CrossFit athletes cannot be solely attributed to morphological changes, indicating that other factors such as neuromuscular adaptations, movement patterns, or psychosocial contributors may be involved in the development of subacromial shoulder pain in this population. Moreover, considering that the observed sonographic difference did not exceed the minimum detectable change, its clinical significance remains limited. Therefore, the utility of USI as a diagnostic tool for unilateral subacromial shoulder pain in symmetrical sports like CrossFit should be interpreted with caution. Still, given the small sample size and the exclusive inclusion of male Rx/Elite athletes, the generalizability of these results to female or recreational populations remains limited.

Thanks for your valuable commentaries which have permitted us to improve the quality of the manuscript.

Sincerely,

The authors

Reviewer 3 Report

Comments and Suggestions for Authors

This manuscript presents an observational study using ultrasound to assess morphological and functional differences between symptomatic and asymptomatic shoulders in male CrossFit participants with unilateral subacromial shoulder pain. While the study follows a standardized protocol and includes thoughtful efforts to minimize operator dependency—such as examiner blinding, use of a single experienced sonographer, and repeated measurements—it ultimately lacks the scientific novelty and clinical impact expected for publication.

The primary finding of the study is a 0.2 mm increase in subacromial deltoid bursa (SADB) thickness in the painful shoulder compared to the non-painful side, which, although statistically significant (p = 0.004), falls below the minimal detectable change (0.31 mm) reported in previous reliability studies. Therefore, the result cannot be considered clinically meaningful. Moreover, none of the other ultrasound parameters, including supraspinatus tendon thickness, acromiohumeral distance, coracoacromial ligament measurements, or muscle thickness differentials, showed significant differences between sides. As a result, the study essentially confirms the absence of notable structural alterations and contributes little to existing knowledge.

Additionally, the term "CrossFit athlete" is used throughout the manuscript, yet the participants were defined only by having more than two years of training experience and exercising at least four times per week. This definition does not meet the conventional standard for competitive or elite athletes and may give readers an inflated impression of the population studied. A more precise description such as “recreational CrossFit practitioners” would be appropriate and should be used to avoid misrepresentation.

The sample size calculation is another concern. Although the authors reference prior literature to justify their target number of participants, the power analysis is based solely on one variable—SADB thickness—without consideration of multiple outcome measures. Furthermore, no pilot data from the target population were used to verify the assumed effect size, and the analysis lacks any statistical correction for multiple comparisons. These factors diminish the methodological robustness of the study.

The conclusion that pain in CrossFit athletes cannot be solely attributed to morphological changes is already well-supported in existing literature and does not offer a new conceptual contribution. Despite the unique training patterns involved in CrossFit, the study fails to demonstrate any specific sonographic adaptations or pathological findings that distinguish this group from other overhead athletes. This significantly limits the originality and relevance of the work.

Finally, the manuscript includes several speculative or overly optimistic statements that suggest potential diagnostic or prognostic value of ultrasound, even though the study design does not support such claims. These statements, while subtle, contribute to an impression of overinterpretation and detract from the scientific objectivity of the manuscript.

In summary, while the technical aspects of the study are adequately handled, the clinical implications are minimal, the novelty is weak, and the use of terminology and statistical justification could be improved. Therefore, I do not believe this manuscript meets the criteria for publication and recommend rejection.

Author Response

Responses to Reviewers’s comments:

Response to Reviewer 3 comment:

In response to “This manuscript presents an observational study using ultrasound to assess morphological and functional differences between symptomatic and asymptomatic shoulders in male CrossFit participants with unilateral subacromial shoulder pain. While the study follows a standardized protocol and includes thoughtful efforts to minimize operator dependency—such as examiner blinding, use of a single experienced sonographer, and repeated measurements—it ultimately lacks the scientific novelty and clinical impact expected for publication.”

  • Thank you for your thoughtful appreciation. While we acknowledge the reviewer’s concerns regarding novelty and impact, we appreciate the recognition of our methodological rigor and the opportunity to contribute data on an understudied athletic population.

In response to “The primary finding of the study is a 0.2 mm increase in subacromial deltoid bursa (SADB) thickness in the painful shoulder compared to the non-painful side, which, although statistically significant (p = 0.004), falls below the minimal detectable change (0.31 mm) reported in previous reliability studies. Therefore, the result cannot be considered clinically meaningful. Moreover, none of the other ultrasound parameters, including supraspinatus tendon thickness, acromiohumeral distance, coracoacromial ligament measurements, or muscle thickness differentials, showed significant differences between sides. As a result, the study essentially confirms the absence of notable structural alterations and contributes little to existing knowledge.”

  • Thank you for your thoughtful comment. We respectfully acknowledge that the main ultrasound finding did not exceed the minimal detectable change and therefore lacks standalone clinical significance. However, we believe that this result, together with the absence of other significant structural differences, offers meaningful insight for clinicians utilizing accessible tools such as musculoskeletal ultrasound. Specifically, it underscores the importance of avoiding overreliance on imaging when interpreting unilateral shoulder pain in symmetrical sports like CrossFit. Although this was not the primary aim of the study, we have introduced specific modifications to the Discussion and Conclusion sections to better convey this perspective and improve the clarity and scientific quality of the manuscript, while remaining within its intended scope.
    • “Discussion: The main purpose of the study was to compare morphology and changes in muscle thickness by ultrasound between the subacromial shoulder pain versus the asymptomatic side of male CrossFit athletes. In summary, our findings showed a statistically significant difference in thickness of the SADB of the painful shoulder compared to the asymptomatic side in CrossFit athletes with SSP. All other ultrasound variables did not show differences between PS and NPS. The main finding of our study was the presence of an increase in the thickness of the SADB of the PS versus the NPS of 0.2 mm. Repetitive impingement associated with movements performed at extreme high intensity and range of motion is associated with soft tissue microtrauma, leading to thickening of collagen fibers and potential space conflict in the shoulder 32. Previous research has shown soft-tissue differences in athletes, reporting that ultrasound might play an interesting role as a prognostic tool in athlete prevention programs 33. To our knowledge, only a few studies have directly evaluated structural alterations in CrossFit athletes using imaging modalities such as MRI or USI. Bernstorff et al. 14 identified frequent bursal and tendinous abnormalities in symptomatic CrossFit athletes, reinforcing the relevance of exploring such findings through more accessible tools like musculoskeletal ultrasound. From a biological perspective, an increased thickness of the SADB in the superior glenohumeral space can be supported with respect to the mechanics of overhead movements. Nonetheless, despite the observed increase in SADB thickness in the PS versus the NPS of 0.20 mm difference, the clinical relevance of these differences cannot be justified based on the reliability study values reported by Kjaer et al. (MDC = 0.31; SEM = 0.08) 23. Thus, although the significant differences (p = 0.004) and the moderate effect size reported (d = 0.74), the difference may not reflect a true physiological change beyond measurement error. This highlights the need for cautious interpretation when translating small morphologic changes into clinical decisions. In this context, SADB thickness should not be used in isolation as a diagnostic marker, and its utility in identifying or monitoring subacromial pain must be integrated with functional, symptomatic, and contextual factors. Several research determined a weak relationship between ultrasound findings and shoulder pain has been described 19,20,34,35. Couanis et al. reported a progressive increase in SADB thickness with greater training duration and volume among open-water marathon swimmers. While this thickening was not associated with pain, it was attributed to an adaptive response to repetitive movement. Pain was only linked to an acute increase in SADB thickness following competition 36. In this context, the detection of structural abnormalities by imaging tests alone is insufficient to elucidate the relationship between pain and local tissue changes 35. However, increased SADB thickness has been proposed as a possible structural adaptation in individuals with SSP that may lead to decreased subacromial space. Miyake et al. reported a reduction in CAL thickness in participants with rotator cuff tears 28. Nevertheless, our results did not report changes between shoulders in the AHD and CAL distance and thickness 27,37,38. A possible explanation of the lack of structural differences could be associated with the fact that CrossFit is related to bilateral overhead movements. Hence, it could be proposed that structural adaptation could be similar in both shoulders, regardless of the presence of symptoms. Our results align with some previous studies reporting bursal thickening in symptomatic populations 14,19, yet other investigations in overhead athletes have failed to find consistent sonographic differences between painful and asymptomatic shoulders 19,20,35. These discrepancies highlight the complex, multifactorial nature of shoulder pain and the necessity of contextualizing structural findings within a broader clinical and functional framework. Otherwise, Siang Ting et al. determined relationships between coracohumeral distance and coracohumeral ligament thickness in patients with SSP 39. Indeed, the coracohumeral ligament plays an important role in stabilizing the glenohumeral head during inferior displacement and external rotation, extreme movements to which the glenohumeral joint is frequently exposed during the execution of overhead weightlifting gestures often performed in CrossFit. Impingement of structures during space-closing mechanisms, such as the subacromial space during head movements, can result in mechanical compromise of the tendon. This leads to increased compressive forces on the bursal tissue, paratenon, or bony periosteum, which are richly innervated by free nerve endings and nociceptors, playing an important role in pain and sensitivity. From this perspective, it has been hypothesized that mechanical stress in overhead athletes may contribute to both peripheral and central sensitization mechanisms 40. CrossFit athletes are exposed to high workloads performed at high intensity, with short rest times and complex sequential gestures that accentuate the mechanical demands placed on the shoulder and may increase the risk of peripheral fatigue and related strength deficits 41. Although our study was not designed to directly evaluate these neurophysiological processes, the observed structural findings, combined with the intensity of CrossFit training, warrant further investigation of these mechanisms in future research; 4.1. Limitations and future research: Some study limitations must be considered, such as the exclusion of women due to the potential risk of bias associated with sex-related differences. Although the sample size was calculated, the findings of this study must be interpreted with caution. Moreover, subgroup analyses by arm dominance were not performed, which could have provided additional insights into unilateral adaptations and their potential relation to shoulder pain. Future studies may focus on the assessment of the bilateral adaptive response of sonographic shoulder morphology in CrossFit athletes with SSP, exploring different shoulder spaces such as the postero-internal corner, the rotator interval, or the subcoracoid space, as well as related soft-tissue structures. Additionally, while quantitative sonographic measurements (e.g., thickness, echogenicity) remain essential, the incorporation of structured grading systems based on expert pattern recognition could provide complementary insights into subtle morphological alterations that may not be captured by linear or pixel-based variables alone. Tools such as the Modified Öhberg scale (based on five items for tendon vascularization) 42, Ultrasound Tissue Characterization (UTC) (for fiber organization and echo categorization) 43, or Likert-type expert-based grading systems, along with their intra- and inter-rater validation using defined criteria for identifying tendon abnormalities (e.g., delamination, disruption of fiber pattern, or the presence of echo defects), as proposed by Yoon K et al.44, may provide a more comprehensive framework for interpreting ultrasound findings in athletes, integrating both structural evaluation and pain-related clinical insight 40,45. These tools could be operationalized in future studies through consensus-based protocols or training systems for experienced examiners. For example, standardized sonographic assessment protocols, examiner training to ensure inter-session reliability, and the use of structured scoring forms or classification templates may facilitate consistent real-time categorization of sonographic findings. Incorporating these systems into future study designs could enhance the clinical interpretation of shoulder ultrasound specifically in male CrossFit athletes by linking sonographic patterns to pain intensity, perceived function, or return-to-training progression. Building upon this structural perspective, future research should also explore the relationship between structural changes in the SADB and structural components of the shoulder joint, as well as in the evaluation of the posterointernal region of the shoulder and structures such as the posterior glenohumeral capsule, the infraspinatus tendon thickness and the posterointernal impingement. This could help clarify whether isolated changes in the SADB reflect a localized phenomenon or form part of a broader pattern of tissue adaptation or dysfunction in overhead athletes. Finally, considering the high physical demands and psychological pressures inherent to competitive CrossFit, it would be valuable to assess pain-related beliefs and psychological constructs such as fear avoidance behaviors, kinesiophobia, pain catastrophizing, or self-efficacy in relation to the presence of pain and structural abnormalities. Their integration into future study designs could contribute to a more comprehensive biopsychosocial understanding of shoulder pain in this athletic population; 5. Conclusion: No sonographic differences were observed between the PS and NPS in CrossFit athletes. However, small differences in SADB thickness were observed. These findings suggest that the presence of pain in CrossFit athletes cannot be solely attributed to morphological changes, indicating that other factors such as neuromuscular adaptations, movement patterns, or psychosocial contributors may be involved in the development of subacromial shoulder pain in this population. Moreover, considering that the observed sonographic difference did not exceed the minimum detectable change, its clinical significance remains limited. Therefore, the utility of USI as a diagnostic tool for unilateral subacromial shoulder pain in symmetrical sports like CrossFit should be interpreted with caution. Still, given the small sample size and the exclusive inclusion of male Rx/Elite athletes, the generalizability of these results to female or recreational populations remains limited.”

In response to “Additionally, the term "CrossFit athlete" is used throughout the manuscript, yet the participants were defined only by having more than two years of training experience and exercising at least four times per week. This definition does not meet the conventional standard for competitive or elite athletes and may give readers an inflated impression of the population studied. A more precise description such as “recreational CrossFit practitioners” would be appropriate and should be used to avoid misrepresentation.”

  • Thank you for your comment. We agree that clear characterization of the study population is essential. To avoid any ambiguity, we have revised the description of participants to specify that all individuals were Rx or Elite athletes, following the official CrossFit performance classification system. This clarification has been added to the inclusion criteria and consistently reflected throughout the manuscript to ensure an accurate representation of the competitive level of the participants.
    • “2.3. Participants: Twenty male (N=20) CrossFit athletes with unilateral shoulder pain were recruited, evaluated bilaterally (n=40) and divided into two groups of 20 PS (n=20) and 20 NPS (n=20) groups. All participants were informed and gave signed consent for participation in the study. Data were collected between January and June 2023 and inclusion criteria for study enrollment were: (a) male CrossFit athletes; (b) aged 18-35; (c) with at least two years of experience in CrossFit discipline; (d) athletes who follow (e) competitive athletes classified in the Rx or Elite category, according to CrossFit’s standardized performance classification system; (f) athletes following a structured and individualized training program, and (g) presenting at least two of the following five shoulder impingement criteria: positive Neer sign, positive Hawkins’s sign, positive Jobe sign, pain with apprehension 24. Additionally, (h) participants had to present pain in overhead weightlifting activities in the last 3 months, such as snatch and jerk variants during training or competition. Participants were excluded if they reported a history of surgical interventions or congenital alterations; metabolic, neurological, autoimmune, or cardiovascular diseases affecting the shoulder; inability to understand the study protocol; cognitive impairments; taking analgesic or anti-inflammatory medication during the study or a week before; or a low physical activity level on the International Physical Activity Questionnaire (IPAQ) in the last week.”
    • “5. Conclusion: No sonographic differences were observed between the PS and NPS in CrossFit athletes. However, small differences in SADB thickness were observed. These findings suggest that the presence of pain in CrossFit athletes cannot be solely attributed to morphological changes, indicating that other factors such as neuromuscular adaptations, movement patterns, or psychosocial contributors may be involved in the development of subacromial shoulder pain in this population. Moreover, considering that the observed sonographic difference did not exceed the minimum detectable change, its clinical significance remains limited. Therefore, the utility of USI as a diagnostic tool for unilateral subacromial shoulder pain in symmetrical sports like CrossFit should be interpreted with caution. Still, given the small sample size and the exclusive inclusion of male Rx/Elite athletes, the generalizability of these results to female or recreational populations remains limited.”

In response to “The sample size calculation is another concern. Although the authors reference prior literature to justify their target number of participants, the power analysis is based solely on one variable—SADB thickness—without consideration of multiple outcome measures. Furthermore, no pilot data from the target population were used to verify the assumed effect size, and the analysis lacks any statistical correction for multiple comparisons. These factors diminish the methodological robustness of the study.”

  • .

In response to “The conclusion that pain in CrossFit athletes cannot be solely attributed to morphological changes is already well-supported in existing literature and does not offer a new conceptual contribution. Despite the unique training patterns involved in CrossFit, the study fails to demonstrate any specific sonographic adaptations or pathological findings that distinguish this group from other overhead athletes. This significantly limits the originality and relevance of the work.”

  • Thank you for your valuable comment. We acknowledge that the sample size calculation was based exclusively on the SADB thickness variable, as it was considered the primary outcome of interest in our study. We agree that the inclusion of additional outcome measures and corrections for multiple comparisons would enhance the methodological robustness. However, given the exploratory nature of the study and the lack of available pilot data from this specific athletic population, we prioritized a focused, conservative estimate using previously published reliability parameters. This limitation has been explicitly acknowledged in the revised Limitations section of the manuscript, and we appreciate your suggestion as a key consideration for future study designs.
    • 1. Limitations and future research: Some study limitations must be considered, such as the exclusion of women due to the potential risk of bias associated with sex-related differences. Although the sample size was calculated, the findings of this study must be interpreted with caution. In particular, the sample size calculation was based solely on SADB thickness as the primary outcome, without corrections for multiple comparisons, which may limit the statistical power to detect differences in other morphological variables.

In response to “Finally, the manuscript includes several speculative or overly optimistic statements that suggest potential diagnostic or prognostic value of ultrasound, even though the study design does not support such claims. These statements, while subtle, contribute to an impression of overinterpretation and detract from the scientific objectivity of the manuscript.

In summary, while the technical aspects of the study are adequately handled, the clinical implications are minimal, the novelty is weak, and the use of terminology and statistical justification could be improved. Therefore, I do not believe this manuscript meets the criteria for publication and recommend rejection.”

  • Thank you for your thoughtful evaluation. We respectfully disagree with the perception that the manuscript includes speculative or overly optimistic claims regarding the diagnostic or prognostic utility of ultrasound. Throughout the text, we have aimed to maintain scientific objectivity, explicitly acknowledging the exploratory nature of our study, the absence of clinically meaningful differences, and the limitations in generalizability. Additionally, we have revised the Discussion and Conclusion sections to ensure that any interpretative remarks are grounded in the data and framed within the broader context of existing literature. We appreciate your perspective and have taken it into account to reinforce the clarity and restraint of our final statements.

Thanks for your valuable commentaries which have permitted us to improve the quality of the manuscript.

Sincerely,

The authors

Round 2

Reviewer 3 Report

Comments and Suggestions for Authors

The authors have adequately addressed the concerns raised during the initial review and substantially improved the quality, clarity, and objectivity of the manuscript. Key limitations such as the marginal clinical significance of the SADB thickness difference, the absence of additional significant ultrasound parameters, and the constrained statistical approach have been transparently acknowledged and discussed. Importantly, the authors have reframed their findings not as a diagnostic breakthrough but as a cautionary interpretation against over-reliance on imaging findings in overhead athletes with unilateral shoulder pain. This shift in perspective enhances the relevance of the study for clinical decision-making.

The definition of the study population has also been appropriately clarified. The original concern regarding the use of the term “CrossFit athlete” has been resolved by specifying that participants were classified as Rx or Elite level according to standardized CrossFit criteria. This correction strengthens the internal consistency and external validity of the manuscript.

Regarding the sample size justification, the authors clearly stated that the calculation was based solely on SADB thickness as the primary outcome, and that adjustments for multiple comparisons were not performed. This limitation is now explicitly recognized in the manuscript, reflecting methodological transparency.

Additionally, earlier concerns about overly speculative or optimistic claims about the diagnostic or prognostic utility of ultrasound have been addressed. The revised discussion and conclusion are more balanced, with cautious interpretation grounded in the data and framed within the broader context of existing literature. The authors emphasize that structural findings alone are insufficient to explain shoulder pain in this population and call for future research to integrate functional and psychosocial dimensions.

While the novelty of the study remains modest, as the structural findings largely align with prior research in other overhead athletes, the focus on Rx/Elite-level CrossFit participants offers value in documenting ultrasound findings within this specific athletic subgroup. The authors’ thoughtful integration of literature, careful revision of language, and improved clarity throughout the manuscript warrant a positive recommendation.

Given the thorough and well-executed revisions, I find the manuscript acceptable for publication in its current form.